# Topographic Differential Diagnosis of Chronic Plaque Psoriasis: Challenges and Tricks

**DOI:** 10.3390/jcm9113594

**Published:** 2020-11-08

**Authors:** Paolo Gisondi, Francesco Bellinato, Giampiero Girolomoni

**Affiliations:** Section of Dermatology and Venereology, Department of Medicine, University of Verona, 37129 Verona, Italy; francesco.bellinato@gmail.com (F.B.); giampiero.girolomoni@univr.it (G.G.)

**Keywords:** psoriasis, plaque psoriasis, differential diagnosis, diagnosis, papulosquamous lesions

## Abstract

Background: Psoriasis is an inflammatory skin disease presenting with erythematous and desquamative plaques with sharply demarcated margins, usually localized on extensor surface areas. Objective: To describe the common differential diagnosis of plaque psoriasis classified according to its topography in the scalp, trunk, extremities, folds (i.e., inverse), genital, palmoplantar, nail, and erythrodermic psoriasis. Methods: A narrative review based on an electronic database was performed including reviews and original articles published until 1 September 2020, assessing the clinical presentations and differential diagnosis for psoriasis. Results: Several differential diagnoses could be considered with other inflammatory, infectious, and/or neoplastic disorders. Topographical differential diagnosis may include seborrheic dermatitis, tinea capitis, lichen planopilaris in the scalp; lupus erythematosus, dermatomyositis, cutaneous T-cell lymphomas, atopic dermatitis, syphilis, tinea corporis, pityriasis rubra pilaris in the trunk and arms; infectious intertrigo in the inguinal and intergluteal folds and eczema and palmoplantar keratoderma in the palms and soles. Conclusions: Diagnosis of psoriasis is usually straightforward but may at times be difficult and challenging. Skin cultures for dermatophytes and/or skin biopsy for histological examination could be required for diagnostic confirmation of plaque psoriasis.

## 1. Introduction

Psoriasis is a chronic inflammatory skin disease affecting 2–3% of the general population [1]. Plaque psoriasis, also known as psoriasis vulgaris, is the most common type of psoriasis, comprising approximately 80 to 90% of cases [2]. Typical psoriatic lesions are erythematous, desquamative, and rounded plaques with sharply demarcated margins [2]. Psoriasis may be localized in any part of the body. The sites of predilection generally include the scalp, elbows, knees, and lower back, with a bilateral and symmetric distribution. Psoriatic lesions may sometimes predominate in seborrheic areas of the face and scalp, a subset known as sebopsoriasis [3]. New lesions may appear at the site of cutaneous trauma from scratching or pressure, known as the Koebner phenomenon [4]. The color of the plaques is a full, rich red (often referred to as “salmon pink”). The amount of scaling is variable, but most lesions are surmounted by a characteristic silvery-white scaling, which may vary considerably in thickness and is easily elicited by gently rubbing the lesions with a sharp object. The removal of psoriatic scales usually reveals an underlying smooth, glossy, red membrane with small bleeding points where the thin suprapapillary epithelium is torn off (Auspitz sign) [5]. Diagnosis is usually clinically easy but may at times be difficult and challenging because the psoriatic lesions could be similar to those of other inflammatory, infectious, and/or neoplastic skin diseases [1]. The aim of this article is to describe the common differential diagnosis of plaque psoriasis classified according to its topography in the scalp, trunk, extremities, folds (i.e., inverse), genital, palmoplantar, nail, and erythrodermic psoriasis. A narrative review based on electronic searches on the PubMed^®^ database was performed. The keywords used were “psoriasis,” “plaque psoriasis,” “differential diagnosis,” “diagnosis,” and “papulosquamous lesions.” Reviews and original articles published up to 1 September 2020, including case reports, assessing the description of the clinical presentations and differential diagnosis for psoriasis, were included.

## 2. Scalp

Scalp involvement is very common in psoriasis and presents with well-defined desquamative plaques, usually extending approximately 1 cm beyond the hairline and advancing to the upper neck, the retroauricular regions, and the face [6]. Scalp psoriasis is a frequent cause of severe itch and discomfort for the patient because scales accumulate visibly on dark clothing. When thick silvery or yellowish scales encircle the hair shafts and bind down tufts of hair, the scales may resemble asbestos. This condition is referred to as pityriasis amiantacea and may be the first sign of psoriasis in children and young adults [7]. Scalp psoriasis does not usually induce alopecia. The most common differential diagnoses of scalp psoriasis include seborrheic dermatitis, tinea capitis, and lichen planopilaris (Figure 1).

Seborrheic dermatitis is defined by patches that vary from pink-yellow to red-brown, surmounted by flaky greasy scales. It predilects the areas rich in sebaceous glands such as the scalp, face, ears, and presternal region [8]. Pityriasis simplex capitis (i.e., dandruff) is defined as a diffuse, slight to moderately fine white or greasy scaling of the scalp and terminal hair-bearing areas of the face (beard area) but without significant erythema. This common disorder may be considered the mildest form of seborrheic dermatitis of the scalp. In seborrheic dermatitis, there is erythema in addition to dandruff. Towards the forehead, erythema and scaling are usually sharply demarcated from uninvolved skin, with the border either at the hairline or slightly transgressing beyond it [8]. Distinction of seborrheic dermatitis from psoriasis could be difficult and there may be an overlap in some patients, such as those with sebopsoriasis. However, the plaques of psoriasis tend to be thicker, with silvery-white scales, more discrete and not associated with seborrhea [3]. When a differential diagnosis is not easily made on a clinical basis, videocapillaroscopy could be a useful noninvasive approach for differentiating between psoriasis and seborrheic dermatitis, especially when the scalp is the only affected site. Scalp psoriasis exhibits homogeneously tortuous and dilated capillaries (bushy pattern) with a larger diameter than seborrheic dermatitis [9].

Tinea capitis is a disease caused by a fungal infection of the skin, hair shafts, and follicles of the scalp. Causative agents of tinea capitis include keratinophilic fungi termed dermatophytes from the species of genera Trichophyton or Microsporum. From the site of inoculation, the fungal hyphae grow centrifugally in the stratum corneum. The fungus continues downward growth into the hair, invading keratin as it is formed. Infected hairs are brittle and broken hairs are evident [10,11]. Clinical presentation of tinea capitis varies from a dry, scaly, and noninflamed dermatosis to an inflammatory disease with scaly erythematous lesions and hair loss that may progress to severely inflamed deep abscesses termed kerion, with the potential for scarring and permanent alopecia. Tinea capitis is usually more circumscribed, less diffuse than seborrheic dermatitis, or psoriasis unassociated with hair loss or broken-off hairs. Tinea capitis occurs almost exclusively in children and is very rare in adults. Laboratory diagnosis of tinea capitis depends on direct microscopic examination and/or the culture of skin scrapings or hair plucking (epilated hair) from lesions. Selected hair samples are cultured or allowed to soften in 10–20% potassium hydroxide (KOH) before examination under a microscope. The culture provides precise identification of the species [12]. Trichoscopy may sometimes be sufficient to establish the diagnosis of tinea capitis because of characteristic trichoscopic features [13].

Lichen planopilaris is a chronic inflammatory disorder characterized by follicular and perifollicular scaly and pruritic papules on the scalp (Figure 2). These lesions usually progress over time to atrophic cicatricial alopecia [14]. This disorder is more common in women than men and may be associated with ungual and erosive mucosal involvement [14]. A variant of lichen planopilaris is frontal fibrosing alopecia. It is characterized by marginal progressive hair loss on the scalp, eyebrows, and axillae. Skin histological examination of the scalp lesions is diagnostic [15]. Common findings include lichenoid lymphocyte infiltration in the follicular dermoepidermal junction, wedge-shaped hypergranulosis, colloid bodies, loss of sebaceous glands, and the destruction of hair follicle root sheaths and follicular plugging [15]. Lichen planopilaris may coexist with psoriasis, albeit rarely [16].

## 3. Trunk

Plaque psoriasis commonly affects the umbilical and lumbosacral areas. The trunk is commonly involved in guttate psoriasis. Guttate psoriasis presents with an acute eruption of round or droplet-like (“gutta”), erythematous, and slightly scaling macules and papules [1]. Guttate psoriasis is common in children and young adults with a family history of psoriasis following an acute infection, such as streptococcal pharyngitis [17]. The major differential diagnoses of psoriasis localized in the trunk include pityriasis rosea, maculopapular drug eruptions, pityriasis rubra pilaris, secondary syphilis, tinea corporis, mycosis fungoides, pityriasis lichenoides chronica, and subacute cutaneous lupus erythematosus (Figure 3) [18].

Pityriasis rosea is a benign, self-limiting, papulosquamous eruption most frequently affecting young adults. The disease typically begins with a solitary, salmon-colored patch that heralds the eruption and is commonly referred to as the “herald patch” [19]. This initial lesion enlarges over a few days with a collarette of a fine scale inside the well-demarcated border [20]. Within the following 1–2 weeks, a generalized exanthem consisting of bilateral and symmetric macules with a collarette scale develops on the trunk. The distribution of the lesions is usually with the long axes running parallel to skin tension lines producing the classic “Christmas tree” pattern. This phase tends to spontaneously resolve over the following 6–8 weeks [19].

Pityriasis rubra pilaris is a chronic inflammatory papulosquamous disorder characterized by reddish-orange scaly plaques, palmoplantar keratoderma showing an orange hue, and keratotic follicular papules commonly seen on the dorsal aspects of the proximal phalanges, the elbows, and the wrists [21]. The disease may progress to erythroderma with distinct areas of uninvolved skin, the so-called islands of sparing, and typically spreads in a craniocaudal direction. Patients first notice redness and scales on the face and the scalp, often followed by redness and thickening of the trunk, limbs, palms, and soles. Nail changes include distal yellow-brown discoloration, subungual hyperkeratosis, longitudinal ridging, nail plate thickening, and splinter hemorrhages. Differential diagnosis with plaque psoriasis may be challenging and is generally supported by histological examination [22].

Syphilis is an infectious venereal disease caused by the spirochete Treponema pallidum. If untreated, syphilis progresses through four stages: primary, secondary, latent, and tertiary [23]. Secondary syphilis may also present with a diffuse papulosquamous eruption (generally nonpruritic and bilaterally symmetrical) distributed on the trunk, proximal extremities, palms, and soles, with generalized nontender lymphadenopathy. It develops within 2–10 weeks after the primary chancre and is most florid 3–4 months after infection. Tiny papular follicular syphilids involving hair follicles may result in patchy alopecia. Reddish-brown papules on the penis or anogenital area can coalesce into large elevated plaques up to 2–3 cm in diameter, known as condylomata lata. From 10 to 15% of patients with secondary syphilis develop superficial mucosal erosions, usually painless, on the palate, pharynx, larynx, glans penis, vulva, or in the anal canal and rectum. Mild constitutional symptoms of malaise, headache, anorexia, aching pains in the bones, and fatigue often are present, as well as fever and neck stiffness [24]. Histological examination could be needed to differentiate syphilis from guttate psoriasis.

Tinea corporis is a superficial dermatophyte infection of the glabrous skin. Tinea corporis typically begins as an erythematous, scaly plaque that may rapidly enlarge. Following central resolution, the lesion may become annular in shape. As a result of the inflammation, scale, crust, papules, and vesicles can develop, especially in the advancing border [12]. A KOH examination of skin scrapings may be diagnostic in tinea corporis. The sample should be taken from the active border of a lesion because this region provides the highest yield of fungal elements. A fungal culture is often used as an adjunct to KOH for diagnosis and is more specific than KOH for detecting a dermatophyte infection; therefore, if the clinical suspicion is high and the KOH result is negative, a fungal culture should be obtained [12]. Moreover, the molecular method of polymerase chain reaction for fungal DNA identification can be applied [25].

Mycosis fungoides is the most common type of cutaneous T-cell lymphoma. Classic mycosis fungoides presents in the three stages of patch, plaque, and tumor. The first stage often proceeds for many years and is characterized by a nonspecific dermatitis, usually with minimal itching. The second stage is characterized by erythematous and finely desquamative, pruritic, plaques that range from 2 cm to more than 20 cm in the greatest diameter. They are not as sharp as psoriasis plaques and tend to vanish at the periphery. Annular or serpiginous patterns with central clearing are common. Patches and plaques may affect any area of the skin, but they are often distributed asymmetrically in sun-protected areas (i.e., lower trunk, hips, buttocks, groin, axillae, and breasts). When mycosis fungoides affects the scalp, it is often accompanied by alopecia [26]. Patch and plaque stages should be differentiated from psoriasis [27]. Histomorphologic and cytomorphologic findings yield clues that lead to a diagnosis in most cases. The demonstration of a dominant T-cell clone in skin biopsy specimens by a molecular assay (i.e., Southern blot, polymerase chain reaction) constitutes an additional diagnostic criterion to distinguish cutaneous T-cell lymphoma from inflammatory dermatoses [26].

Pityriasis lichenoides is a rare inflammatory papulosquamous skin disorder. Pityriasis lichenoides encompasses a spectrum of clinical presentations ranging from acute papular lesions that rapidly evolve into pseudovesicles and central necrosis (pityriasis lichenoides et varioliformis acuta (PLEVA)) to small, scaling, benign-appearing papules (pityriasis lichenoides chronica) [28]. Pityriasis lichenoides chronica presents as small erythematous to reddish-brown scattered and discrete papules. Lesions may be distributed symmetrically or asymmetrically on the trunk, buttocks, and proximal extremities, with occasional acral involvement. Lesions may appear on the palms, soles, face, and scalp. A fine scale is usually found, which has been likened to frosted glass. The eruption often is polymorphic, with lesions at different stages of evolution. Histological examination could be needed to confirm the diagnosis [29].

Subacute cutaneous lupus erythematosus (SCLE) is a nonscarring, photosensitive dermatosis. Lesions typically occur in a photosensitive distribution, typically in the V region of the neck, upper trunk, shoulders, and extensor aspects of the upper arms and generally do not extend below the waist. SCLE is classified into two forms: annular/polycyclic or psoriasiform/papulosquamous. Lesions begin as erythematous papules that evolve into either annular polycyclic or psoriasiform plaques [30]. Sun exposure results in an exacerbation of their disease. Patients may complain of mild pruritus [31]. Papulosquamous form may closely mimic psoriasis or lichen planus. In most cases, clinical, serologic testing, and histologic findings are needed to confirm the diagnosis of SCLE. Characteristic histopathologic findings include vacuolar alteration of the basal cell layer and a subepidermal inflammatory cell infiltrate (usually lymphocytic) that could also distribute around vessels and appendiceal structures; an abundance of mucin often is seen within the dermis [32].

## 4. Extremities

Plaque psoriasis symmetrically affects the elbows and knees in most of the patients [1,2]. The most common differential diagnoses of psoriasis localized in the upper or lower limbs include lichen planus, atopic dermatitis, nummular eczema, pityriasis lichenoides chronica, and dermatomyositis.

Lichen planus is a pruritic papulosquamous eruption usually localized in the extremities, but it may also affect the scalp, genitalia, mucous membranes, and nails. The papules are violaceous, shiny, and polygonal, varying in size from 1 mm to greater than 10 mm in diameter. They can be discrete or arranged in groups of lines or circles. Characteristic fine, white lines, called Wickham striae, are often found on the papules [33]. Hypertrophic lichen planus produces extremely pruritic and verrucous plaque most often found on the extensor surfaces of the lower extremities, especially around the ankles [34]. Nail lesions include longitudinal grooving and ridging, hyperpigmentation, subungual hyperkeratosis, onycholysis, and longitudinal melanonychia. Inflammation rarely results in the permanent destruction of the nail matrix with subsequent pterygium formation [33].

Atopic dermatitis (AD) is a common chronic, pruritic inflammatory skin disorder usually starting in early infancy, also affecting adults [35]. Essential features include pruritus and skin lesions of eczema (acute, chronic) with typical morphology and distribution. In adults, the face, neck, upper trunk, and flexural aspects of the elbows and knees are commonly involved [36]. Acute eczema lesions are vesicular and exudative, and excoriations and crusting are common. Lichenification and desquamation are characteristic of chronic eczema. The skin is usually flaky, rough, and xerotic [37]. Personal and/or family history of atopy, as well as the association with asthma and allergic rhinitis, are common. Itch in AD can be very severe and interfere markedly with sleeping and work/studying ability [35].

Nummular eczema is characterized by very pruritic, coin-shaped lesions. Lesions are exudative and oozing and then become crusted and scaling, resembling psoriasis. It most commonly occurs on the lower limbs. The hands and trunk are frequently affected with from a few to many lesions. Many cases of nummular eczema are phenotypic variants of atopic eczema [38].

Dermatomyositis can easily simulate psoriasis presenting with erythematous–violaceous and scaly lesions on the scalp, elbows, and knees. When dermatomyositis is suspected, Gottron’s papules, heliotrope rash, dystrophic cuticles, and nailfold capillary abnormalities should be investigated [30]. In most cases, clinical, serologic testing, and histologic findings are needed to confirm the diagnosis of dermatomyositis.

## 5. Folds (Inverse Psoriasis)

Inverse or flexural psoriasis is characterized by shiny, smooth, well-defined, pink to red plaques, which are usually not scaling because of the moisture in the fold. Common sites include axillae, submammary folds, the groin, and intergluteal cleft. Flexural psoriasis is rarer than plaque psoriasis and is estimated to affect 3 to 36% of patients [39]. In young infants, it may often present in the diaper area, with a typical involvement of the inguinal folds, also known as napkin psoriasis [40]. The most common differential diagnoses include bacterial, mycotic or Candida intertrigo (i.e., infection of the folds), seborrheic dermatitis, contact dermatitis, and Hailey–Hailey disease (Figure 1).

Erythrasma is a chronic superficial bacterial intertrigo by Corynebacterium minutissimum. The typical appearance of erythrasma is well-demarcated, brown-red macular patches. The skin has a wrinkled appearance with fine scales. Infection is commonly located on the inner thighs, crural region, scrotum, and axillae. Wood light examination of erythrasma lesions reveals the coral-red fluorescence of lesions. The cause of this color fluorescence has been attributed to excess coproporphyrin III synthesis by these organisms, which accumulates in cutaneous tissue and emits a coral-red fluorescence when exposed to a Wood light. A direct Gram-staining examination and culture could also support the diagnosis [41].

Tinea cruris is a pruritic superficial fungal infection of the groin and adjacent skin by dermatophytes such as Trichophyton rubrum and Epidermophyton floccosum. Large patches of erythema with a central clearing are centered on the inguinal creases and extend distally down the medial aspects of the thighs and proximally to the lower abdomen and pubic area. Scales are demarcated sharply at the periphery. Chronic infections typically are dry with a popular–annular or arciform border and barely perceptible scales at the margin. Microscopic examination of a KOH wet mount of scales and scale culture is diagnostic in tinea cruris [12].

Candida intertrigo occurs in skin folds where occlusion (by clothing or shoes) produces abnormally moist conditions. It usually presents as erythema, maceration, and satellite pustules in the folds, accompanied by soreness and pruritus. KOH examination of skin scrapings is the easiest and most cost-effective method for diagnosing cutaneous candidiasis. Culture from an intact pustule skin biopsy tissue can help to support the diagnosis [42,43].

Seborrheic dermatitis may rarely manifest with non-scaling intertrigo of the umbilicus, axillae, inframammary, and inguinal folds. Perineum or anogenital crease may also be involved, particularly in infants and children [44].

Contact dermatitis is an acute or chronic skin inflammation caused by cutaneous interaction with a chemical, biologic, or physical agent. Acute dermatitis is manifested by redness, erythema, mild edema, and vesiculation (oozing). Chronic dermatitis presents with lichenification, hyperkeratotic scale, fissures, or ulcerations. Irritants or allergens that may cause intertrigo include body fluids such as sweat, urine, friction due to movement and clothing, dryness due to antiperspirant and/or soap, excessive washing, fragrances, preservatives in deodorant, and components in underwear (e.g., rubber in the elastic). A patch test could be helpful in identifying allergens responsible for contact dermatitis [45,46].

Hailey–Hailey disease (or familial benign pemphigus) is a rare and chronic autosomal dominant disorder with incomplete penetrance. Vesicles and erythematous-fissured plaques with overlying crusts typically occur in the axillary, inguinal, and genital area, as well as the chest and neck. Burning and itching accompany the eruption, and malodorous drainage occurs in some cases because of secondary infection from bacteria or yeast overgrowth. Multiple asymptomatic longitudinal white bands on the fingernails have also been described. The involvement of mucosa is rare. The characteristic clinical appearance of familial benign pemphigus, as well as biopsy findings revealing intraepidermal and suprabasilar acantholysis, could confirm the diagnosis [47].

## 6. Genital Psoriasis

Genital psoriasis affects up to 60% of patients during their lifetime [48] but is often overlooked by physicians. It can have a significant impact on patients’ psychosocial function due to intrusive physical symptoms, such as genital itch and pain, and a detrimental impact on sexual health and impaired relationships [49]. As in flexural psoriasis, scaling may be mild or absent. Painful fissures of the intergluteal cleft and an intense redness may be a diagnostic clue. In women, hair-bearing regions such as mons pubis and labia majora are commonly affected, sparing the labia minora. Vulvar psoriasis requires differential diagnosis with atopic or contact dermatitis, lichen sclerosus, lichen ruber planus, and vulvar (pre)malignant lesions [50]. In men, both glans and penis shaft may be involved, presenting with well-defined erythematous plaques. Irritative balanitis, Zoon’s balanitis, erythroplasia of Queyrat, or extramammary Paget may mimic genital psoriasis [51]. Either for male and female anogenital dermatosis, a skin biopsy may be helpful to establish a diagnosis when the typical lesions elsewhere on the body or psoriatic nail changes are absent.

## 7. Palmoplantar Psoriasis

Palmoplantar psoriasis (PPP) is a variant of psoriasis that characteristically affects the skin of the palms and soles. PPP is clinically characterized by hyperkeratotic and erythematous plaques. Additional findings include painful fissures and nail involvement (Figure 4) [52]. It can be found either with or without the involvement of other cutaneous regions. PPP should be distinguished from palmoplantar pustulosis, which is characterized by hyperkeratosis and clusters of pustules on an erythematous–desquamative base over the palms and soles. Pustules evolve in brown and yellow crusts at later stages. Palmoplantar pustulosis is more frequent in smoking women and may also be part of synovitis, acne, pustulosis, hyperostosis, and osteitis (SAPHO) syndrome, presenting with a typical involvement of the anterior chest wall joints. The differential diagnosis for PPP includes chronic eczema (either atopic or contact), pompholyx, tinea manuum/pedis, pityriasis rubra pilaris, scabies, palmoplantar pustulosis and, more rarely, hereditary palmoplantar keratoderma [53]. The main differential diagnosis is with chronic hand eczema, where even histology may not be helpful given that psoriasiform epidermal hyperplasia is common in chronic hand eczema and spongiosis may occur in palmoplantar psoriasis. History of vesicles and oozing lesions is very important, as psoriasis is never oozing. In pompholyx, pruritic firm vesicles are found on the palmar surfaces and lateral aspects of the fingers. Generally, the content of the vesicles is clear; however, in the case of superinfection, they may become purulent. Fungal infections are commonly asymmetric; very evocative is the involvement of both feet with only one hand [54]. Microscopic examination of a KOH wet mount of scales and scale culture is diagnostic in fungal infection. Scabies may present acral pustules in infants simulating palmoplantar pustulosis, while crusted scabies may cause palmoplantar hyperkeratosis as PPP. Scabies is very pruritic and accompanied by burrows that may be confirmed through dermoscopy or microscopy from skin scraping. When suspected, a family history for hereditary palmoplantar keratoderma should be investigated.

## 8. Nail Psoriasis

Nail alterations could be found in up to 80–90% of patients with psoriasis or psoriatic arthritis (PsA) in their lifetime and sometimes they are the only manifestation of psoriasis, fetching the diagnosis of the disease [55]. Most patients with psoriatic arthritis present with involvement of distal interphalangeal joints. Nail psoriasis is considered an independent predictor for the development of psoriatic arthritis in patients with psoriasis [56]. Nail psoriasis can manifest clinically as a wide variety of nail changes, as a result of the location of the disease within the nail bed or matrix, including pitting, salmon patches (“oil drop discoloration sign”), onycholysis, splinter hemorrhages, red macules of the lunula, psoriatic white spots, subungual hyperkeratosis, and nail plate thickening. Pitting presents with irregularly distributed depressions over the nail plate. Salmon patches are irregular yellow-orange areas beneath the nail plate. Onycholysis in psoriasis is a distal detachment of the nail plate surrounded by an erythematous border [57]. The psoriatic nail may have more than one clinical manifestation in a single nail depending on the part of the nail apparatus affected. The most common differential diagnoses of nail psoriasis include onychomycosis, traumatic onycholysis, or nail alteration associated with other inflammatory skin disorders such as alopecia areata, lichen planus, chronic dermatitis, or pityriasis rubra pilaris (Figure 1).

The changes of onychomycosis resemble nail psoriasis, and it is sometimes difficult to distinguish between the two. However, nail changes in psoriasis tend to be well demarcated compared to fungal infection. The support of fungal culture, nail clippings with periodic acid Schiff and KOH preparations is very important, but how the scale material is collected is crucial. A clue for traumatic onycholysis is the lack of an erythematous border surrounding the onycholysis. Alopecia areata usually appears as linear ridging, nail pitting, longitudinal nail fissuring, along with nonscarring patchy alopecia of the scalp or other body areas. In lichen planus, nail involvement presents as the thinning of nails with ridges and grooves of the nail plate. Scarring of the cuticle sometimes occurs, leading to pterygium formation. Lichen planus also involves the mucosa or skin.

## 9. Erythroderma

Erythroderma is clinically defined as erythema and scaling involving more than 80% of the body surface area. Additional findings include hair loss and onycholysis. Systemic symptoms include malaise, fatigue, anorexia, fever, and chills. Patients with erythroderma may develop lymphadenopathy, hepatomegaly, splenomegaly, and electrolyte abnormalities due to increased transepidermal water loss. Cardiac failure may occur in patients with pre-existing heart conditions [58]. Erythroderma is a reaction pattern of the skin that can occur in the setting of several different skin disorders, most commonly including psoriasis, dermatitis, lymphoma, drug reactions, and pityriasis rubra pilaris [59]. The development of erythrodermic psoriasis can be triggered by the withdrawal of systemic corticosteroid treatment, erroneous topical therapy, and chloroquine or lithium administration. Clinical clues for psoriatic erythroderma include a pre-existing history of plaque or pustular psoriasis, nail psoriasis, psoriatic arthritis and the sparing of the central face. Skin biopsy for histological examination is needed for differential diagnosis.

## 10. Discussion

Diagnosis of psoriasis is usually straightforward but may be at times difficult and challenging because some clinical features, such as erythema and scaling, are also observed in other skin disorders. Moreover, psoriasis may coexist in the same patient with other skin diseases presenting clinical overlapping features, such as atopic dermatitis, because they are not mutually exclusive diagnoses. Diagnostic doubts may also arise in atypical cases or when the psoriatic lesions are localized in particular sites such as nails, folds, or the genital area. Punch biopsy represents the gold standard for most differential diagnosis, apart from nail psoriasis. A skin biopsy may be required for cases in which the presentation is not typical. Skin histopathology shows a psoriasiform reaction pattern, defined as the presence of epidermal hyperplasia with a regular elongation of the rete ridges with bulbous enlargement of their tips. Dermal papillae contain dilated, congested, and tortuous capillaries. Suprapapillary plates are markedly thinned. The infiltrate is superficial, perivascular, initially predominantly lymphocytic, and later also neutrophilic. Parakeratosis is initially focal and later confluent, containing typical collections of neutrophils (Munro’s microabscesses) [60]. Dermoscopy is a noninvasive in-office method helpful to confirm different inflammatory dermatosis. In psoriasis lesions, 10-fold magnification dermoscopy shows red dots or globules in a regular distribution on a light red or pink background covered with white scales. The vascular features appear as bushy (glomerular) vessels at more detailed magnification (digital dermoscopy) (Figure 5) [9].

The proper diagnosis of psoriasis is also very important for rheumatologists. Psoriasis is associated with PsA in approximately one-third of patients. The onset of psoriasis precedes that of PsA in most cases. Both diseases occasionally occur at the same time or, in rare cases, PsA precedes psoriasis [61]. To confirm a proper diagnosis of PsA, it is important to identify psoriasis distinguishing it from other cutaneous diseases that may mimic it. In addition, some rheumatologic diseases show cutaneous manifestation that may resemble psoriasis, such as subacute cutaneous lupus erythematosus and dermatomyositis.

This narrative review has some limitations. We considered in our review the most common differential diagnosis of psoriasis in clinical practice, without mentioning other uncommon skin disorders presenting with papulosquamous lesions such as porokeratosis or lupus vulgaris. Another limitation of the study is that we focused only on plaque psoriasis, without considering the pustular variant and its differential diagnoses. The strength of this review is that topographical differential diagnosis of psoriasis has not been previously presented in the literature, and this classification could be helpful also for non-dermatologists.

## Figures and Tables

**Figure 1 jcm-09-03594-f001:**
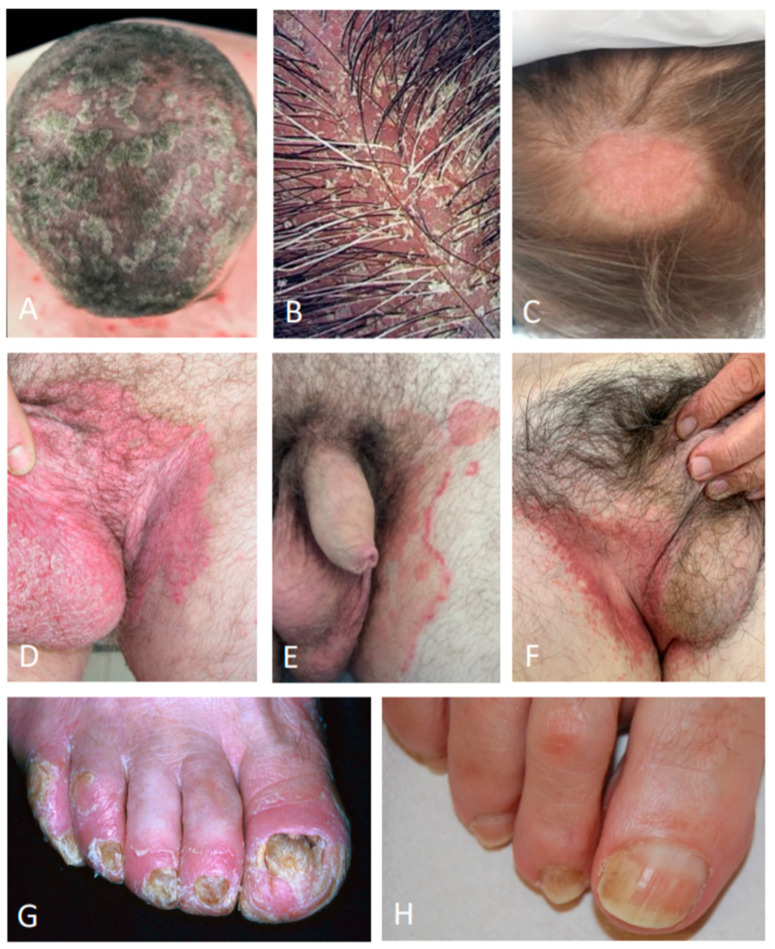
Differential diagnosis of the scalp psoriasis (**A**); seborrheic dermatitis (**B**); tinea capitis (**C**); inverse psoriasis of the inguinal area (**D**); tinea cruris (**E**); candidiasis (**F**); nail psoriasis (**G**); traumatic onycholysis (**H**).

**Figure 2 jcm-09-03594-f002:**
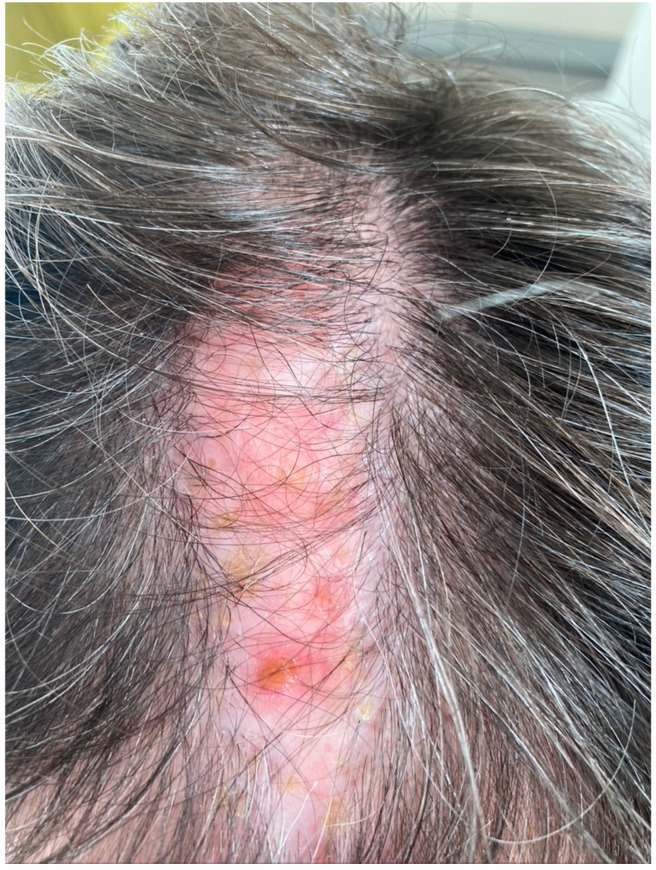
Lichen planopilaris of the scalp.

**Figure 3 jcm-09-03594-f003:**
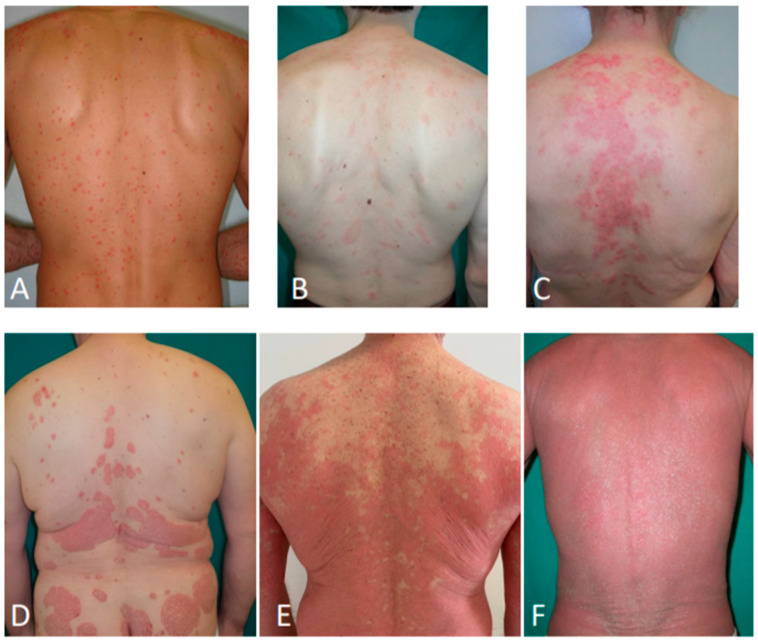
Secondary syphilis (**A**); pityriasis rosea (**B**); subacute cutaneous lupus erythematosus (**C**); psoriasis (**D**); pityriasis rubra pilaris (**E**); erythrodermic mycosis fungoides (**F**).

**Figure 4 jcm-09-03594-f004:**
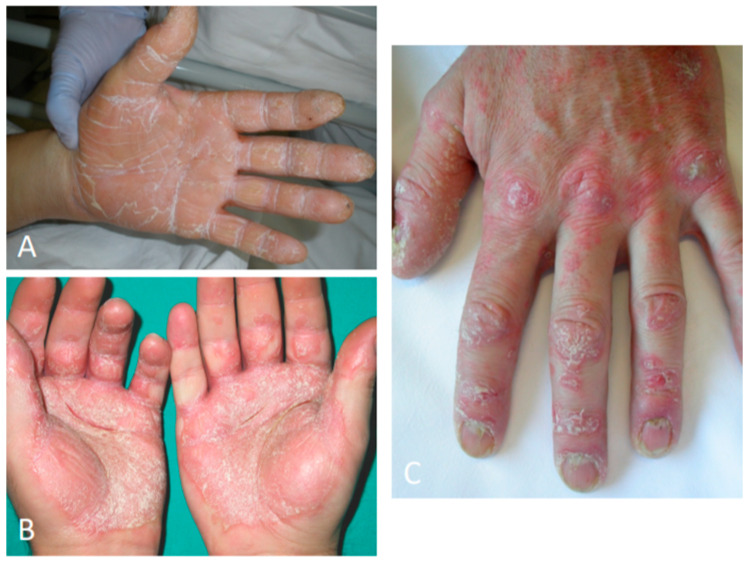
Palmoplantar psoriasis presenting with hyperkeratotic and erythematous plaques of the palmar surfaces of the hands (**A**), fissures (**B**), and digital and nail involvement (**C**).

**Figure 5 jcm-09-03594-f005:**
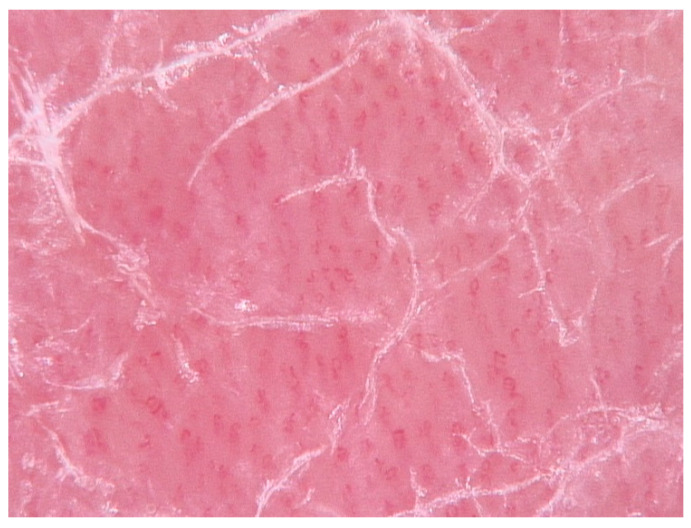
Dermoscopy of a typical psoriatic plaque showing the presence of dilated, elongated, bushy vessels which are homogeneously distributed on a reddish background.

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
