# Peer review of "Topographic Differential Diagnosis of Chronic Plaque Psoriasis: Challenges and Tricks"

_jcm, 2020, doi:10.3390/jcm9113594_

Round 1

Reviewer 1 Report

Interesting paper.

But some revise should be needed.

#1 PPP is a difficult disease, some dermatologist mention "pustulosis palmaris et plantaris", which is different disease from plamoplantar psoriasis.

I think you should prefer this PPP.

And PPP should be distinguish from pomholyx, the authors should be mentioned about more detail differential diagnosis of PPP.

#2 "Scabies" is differential diagnosis for palmoplantar psoriasis/pustulosis.

#3 Recently, dermoscopy is known as auseful tools for dermatologists, even though infllammatory skin disease. The authors should show some dermascopy images.

I am glad that my comment is helpful for you.

Author Response

Re. jcm-966301 - Topographic differential diagnosis of chronic plaque psoriasis:challenges and tricks

Point-by-point response to referee #1

  1. PPP is a difficult disease, some dermatologist mention "pustulosis palmaris et plantaris", which is different disease from palmoplantar psoriasis. I think you should prefer this PPP.
  2. We agree with the reviewer, PPP represents a difficult disease as it might be confused with "pustulosis palmaris et plantaris", which is different disease. Clinical features of pustulosis palmaris et plantaris have been added to the text when we discuss the differential diagnosis of palmoplantar psoriasis. In particular, we included these sentences: “PPP should be distinguished from palmoplantar pustulosis that is characterized by hyperkeratosis and clusters of pustules on an erythematous-desquamative base over the palms and soles. Pustules evolve in brown and yellow crusts at later stages. Palmoplantar pustulosis is more frequent in smoking women and may be also part of the SAPHO syndrome (synovitis, acne, pustulosis, hyperostosis and osteitis) presenting with a typical involvement of the anterior chest wall joints. The differential diagnosis for PPP includes chronic hand eczema (either atopic or contact), pompholyx, tinea manuum/pedis, pityriasis rubra pilaris, scabies, palmoplantar pustulosis and more rarely, hereditary palmoplantar keratoderma.”
  3. And PPP should be distinguished from pompholyx, the authors should be mentioned about more detail differential diagnosis of PPP.
  4. We recognize that pompholyx may simulate PPP, especially when superinfected, and we have included pompholyx description when we discuss the differential diagnosis of PPP. In particular, we added this text: “History of vesicles and oozing lesions is very important, as psoriasis is never oozing. In pompholyx pruritic firm vesicles are found on the palmar surfaces and lateral aspects of the fingers. Generally, the content of the vesicles is clear, however in case of superinfection they may become purulent.”
  5. "Scabies" is differential diagnosis for palmoplantar psoriasis/pustulosis.
  6. We appreciate the suggestion to consider also scabies in the differential diagnosis of palmoplantar psoriasis/pustulosis, as in certain setting it may present either hyperkeratosis (crusted scabies) and pustules (scabies in infants). Consequently, we have included these sentences: “Scabies may present acral pustules in infants simulating palmoplantar pustulosis, while crusted scabies may cause palmoplantar hyperkeratosis as PPP. Scabies is very pruritic and accompanied by burrows that may be confirmed through dermoscopy or microscopy from skin scraping.”
  7. Recently, dermoscopy is known as a useful tool for dermatologists, even though infllammatory skin disease. The authors should show some dermascopy images.
  8. We agree that today dermoscopy is a useful tool for the diagnosis inflammatory skin diseases such as psoriasis. Indeed, a dermoscopy picture of a classical psoriatic plaque and its legend have been included in the manuscript, as Figure 5.

Reviewer 2 Report

the paper i well written

i have few comments

Why hasn’t the authors recommended punch biopsies for determination of diagnosis for the different differential diagnosis?

It can be used in almost all cases of questions of the diagnosis apart from nail psoriasis.

Scalp psoriasis usually extends app one centimetre beyond the hairline.

A figure with lichen plano-pilaris would be helpful.

Figure 2. there is something wrong with the different pictures. There are two D and no F in the legend.

Author Response

Re. jcm-966301 - Topographic differential diagnosis of chronic plaque psoriasis: challenges and tricks

Point-by-point response to referee #2

  1. Why hasn’t the authors recommended punch biopsies for determination of diagnosis for the different differential diagnosis? It can be used in almost all cases of questions of the diagnosis apart from nail psoriasis.
  2. We agree that punch biopsy represents the gold standard for most of the differential diagnosis apart from nail psoriasis. Consequently, we changed the text in the discussion accordingly. In particular, “Punch biopsy represents the gold standard for most of the differential diagnosis, apart from nail psoriasis. A skin biopsy may be required for cases in which the presentation is not typical. Skin histopathology show a psoriasiform reaction pattern, defined as the presence of epidermal hyperplasia with a regular elongation of the rete ridges with bulbous enlargement of their tips. Dermal papillae contain dilatated, congested and tortuous capillaries and suprapapillary plates are markedly thinned. The infiltrate is superficial, perivascular, initially predominantly lymphocytic, and later also neutrophilic. Parakeratosis is initially focal and later confluent, containing typical collections of neuthrophils (Munro’s microabscesses).
  3. Scalp psoriasis usually extends app one centimetre beyond the hairline.
  4. We appreciate the comment and included it in the section describing scalp psoriasis. In particular, “Scalp involvement is very common in psoriasis and presents with well-defined desquamative plaques usually extending approximatively one centimetre beyond the hairline and advancing on the upper neck, the retro-auricular regions and the face.6
  5. A figure with lichen plano-pilaris would be helpful.
  6. Thank you for the suggestion to include also a clinical figure describing lichen plano-pilaris that has been added as Figure 2 along with its legend.
  7. Figure 2. there is something wrong with the different pictures. There are two D and no F in the legend.
  8. Thank you for this comment. The mistake has been corrected.

Round 2

Reviewer 1 Report

This second version of the paper is a great improvement, the authors are to be commended.

#1 For us dermatologist, including you, we know "erythema" is only one erythema, there are so many erythema"s". Not only erythema, psoriasis skin eruption have so many skin phenomenon: large erythema, small erythema, scale, vesicle dot, sometimes small pustule(even though plaque-type-psoriasis). For example, scalp scale is one of koebner phenomenon. 

Therefore, I hope that the authors should mention about/focus strongly on "psoriasis-specific" skin topography for differential diagnosis.
To observe skin eruption is a basic, an important and a daily work for us. That's why I wish your paper should contain serendipity.   

#2 As an other reviewer mentioned, histology or biopsy should be mentioned , I think, even though your title is "Topographic differential diagnosis". histology and topograpy are two sides of the same coin.   

 I hope these comments will be helpful.

Author Response

Verona, 03 November 2020

Prof. Emmanuel Andrès

Chief Editor
Journal of Clinical Medicine

Dr Alen Zabotti

Guest Editor

Journal of Clinical Medicine

Re. jcm-966301 R1- Topographic differential diagnosis of chronic plaque psoriasis: challenges and tricks

Dear Prof. Emmanuel Andrès,

Dear Alen,

We respectfully submit an amended version of the above manuscript for publication consideration in the special issue of Journal of Clinical Medicine entitled “The management of psoriatic arthritis: interface between rheumatology and dermatology, early diagnosis and monitoring of disease activity.”

Please find below a point-by-point response to the reviewer comments. We thank the reviewers for the constructive comments that have improved the manuscript.

We shall look forward to hearing from you.

Kindest regards

Paolo

Prof. Paolo Gisondi

University of Verona

Department of Medicine

Section of Dermatology and Venereology

Re. jcm-966301 R1 - Topographic differential diagnosis of chronic plaque psoriasis: challenges and tricks

 Point-by-point response to reviewer #1

1) COMMENTS TO THE AUTHORS

This second version of the paper is a great improvement, the authors are to be commended.For us dermatologist, including you, we know "erythema" is only one erythema, there are so many erythema"s". Not only erythema, psoriasis skin eruption have so many skin phenomenon: large erythema, small erythema, scale, vesicle dot, sometimes small pustule (even though plaque-type-psoriasis). For example, scalp scale is one of koebner phenomenon. Therefore, I hope that the authors should mention about/focus strongly on "psoriasis-specific" skin topography for differential diagnosis. To observe skin eruption is a basic, an important and a daily work for us. That's why I wish your paper should contain serendipity.

REPLY TO THE REVIEWER 1

We thank the reviewer for the supportive comment. We agree with the reviewer that psoriasis is a papulo-squamous inflammatory skin disorder that should be distinguished from several skin diseases presenting with papulo-squamous lesions. This differential diagnosis is difficult in daily practice because there may be several different degrees of erythema from faint to brilliant and also many degrees of desquamation from minimal to thick and heavy scales. In this manuscript we are suggesting readers to address psoriasis specific skin topography, as an additional clinical element to be considered in the diagnostic approach to patient.

2) COMMENTS TO THE AUTHORS

As another reviewer mentioned, histology or biopsy should be mentioned, I think, even though your title is “Topographic differential diagnosis” histology and topograpy are two sides of the same coin.

REPLY TO THE REVIEWER 1

We fully agree with the Reviewer about the importance of considering skin biopsy and histology for the correct differential diagnosis of psoriasis. Indeed, in the discussion we state: “Punch biopsy represents the gold standard for most of the differential diagnosis of psoriasis, apart from nail psoriasis. A skin biopsy may be required for cases in which the presentation is not typical. Skin histopathology show a psoriasiform reaction pattern, defined as the presence of epidermal hyperplasia with a regular elongation of the rete ridges with bulbous enlargement of their tips. Dermal papillae contain dilatated, congested and tortuous capillaries and suprapapillary plates are markedly thinned. The infiltrate is superficial, perivascular, initially predominantly lymphocytic, and later also neutrophilic. Parakeratosis is initially focal and later confluent, containing typical collections of neuthrophils (Munro’s microabscesses)”.

Reviewer 2 Report

The authors has taken all the comments in to account.

Author Response

Re. jcm-966301 R1- Topographic differential diagnosis of chronic plaque psoriasis: challenges and tricks

Point-by-point response to reviewer #2

1) COMMENTS TO THE AUTHORS

The authors have taken all the comments into account.

REPLY TO THE REVIEWER 2

We thank very much the Reviewer for the supportive comment and for having helped us in improving the manuscript.